# Investigation of a UK biobank cohort reveals causal associations of self-reported walking pace with telomere length

Paddy C. Dempsey [1,2,3,4✉], Crispin Musicha[2,5], Alex V. Rowlands[1,2], Melanie Davies[1,2,6], Kamlesh Khunti[7,8], Cameron Razieh [1,2,8], Iain Timmins [6], Francesco Zaccardi[7,8], Veryan Codd[2,9], Christopher P. Nelson[2,9], Tom Yates[1,2,10] & Nilesh J. Samani[2,9,10]

Walking pace is a simple and functional form of movement and a strong predictor of health status, but the nature of its association with leucocyte telomere length (LTL) is unclear. Here we investigate whether walking pace is associated with LTL, which is causally associated with several chronic diseases and has been proposed as a marker of biological age. Analyses were conducted in 405,981 UK Biobank participants. We show that steady/average and brisk walkers had significantly longer LTL compared with slow walkers, with accelerometer-assessed measures of physical activity further supporting this through an association between LTL and habitual activity intensity, but not with total amount of activity. Bi-directional mendelian randomisation analyses suggest a causal link between walking pace and LTL, but not the other way around. A faster walking pace may be causally associated with longer LTL, which could help explain some of the beneficial effects of brisk walking on health status. Given its simple measurement and low heritability, self-reported walking pace may be a pragmatic target for interventions.

---

[1] Diabetes Research Centre, University of Leicester, Leicester General Hospital, Leicester, UK. [2] NIHR Leicester Biomedical Research Centre, Leicester General and Glenfield Hospitals, Leicester, UK. [3] Baker Heart and Diabetes Institute, Melbourne, VIC, Australia. [4] MRC Epidemiology Unit, University of Cambridge, Cambridge, UK. [5] Department of Health Sciences, University of Leicester, Leicester, UK. [6] Leicester Diabetes Centre, Leicester General Hospital, University Hospitals of Leicester NHS Trust, Leicester, UK. [7] NIHR Collaboration for Leadership in Applied Health Research and Care—East Midlands, University as Leicester, Leicester, UK. [8] Leicester Real World Evidence Unit, University of Leicester, Leicester General Hospital, Leicester, UK. [9] Department of Cardiovascular Sciences, University of Leicester, Leicester, UK. [10]These authors contributed equally: Tom Yates, Nilesh J. Samani.
✉email: pcd5@leicester.ac.uk

 1

Walking is a simple and largely accessible form of physical activity for all ages, conferring many physical, mental, and social health benefits with minimal adverse effects[1–4]. It therefore holds potential as a pragmatic target for intervention[5]. Strong associations with health status have been seen for habitual or self-rated walking pace, which has been associated with better physical fitness and reduced risk of cardiovascular disease and all-cause mortality[6–10], with brisk walkers having up to 20 years greater life expectancy compared to slow walkers[11]. Indeed, walking pace has been shown to have a stronger association with survival and be a substantially better prognostic marker for all-cause or cardiovascular mortality than other measures of physical activity volume, diet, or physical function[12,13]. Similarly, accelerometer-assessed measures of physical activity in UK Biobank suggest that as little as 10 minutes of brisk walking a day is associated with longer life expectancy[14]. A genome-wide association study (GWAS) on self-reported walking pace within UK Biobank identified 70 independent SNPs at genome-wide significance[15], close to an order of magnitude greater than the number reported for other self-reported or accelerometer-assessed measures of physical activity traits within the same cohort[16].

The importance of walking pace as a marker and potential promoter of health is likely to reflect it being a complex functional activity influenced by many factors, such as motor control, musculoskeletal health, cardiorespiratory fitness and lung capacity, habitual activity levels, cognition, motivation, and mental health[3,13,15]. These factors also align with the concept of biological age, which relates to an individual's ability to maintain a robust homoeostasis when subject to stressors[17]. Therefore, it is possible that walking pace acts as both a marker and modulator of biological age. However, whether walking pace is causally associated with potential indicators of biological age remains unknown.

Although the relationship between leucocyte telomere length (LTL) and disease is complex[18], LTL has been proposed as a marker of biological age and is associated with higher risk of several age-related diseases; including coronary artery disease and several cancers[18–23]. Telomeres are DNA–protein complexes that protect the ends of chromosomes from degradation, end-to-end fusion, and abnormal recombination of DNA strands (genomic instability). The DNA component progressively shortens with each cell cycle, decreasing in most cell types as humans age, ultimately contributing to replicative senescence[19,24]. Along with reflecting cellular replicative history, telomere shortening is also moderated by factors such as oxidative stress and inflammation[24]. Telomere length is usually measured in leucocytes (LTL), which is reflective of telomere length in other tissues, along with reflecting the senescent status of circulating cells related to the immune system[25].

Previous research suggests an association of higher levels of physical activity and cardiorespiratory fitness with longer LTL[26,27], supporting the hypothesis that higher levels of physical activity and cardiorespiratory fitness may act to slow markers of biological ageing. However, most human studies to date have been small and/or observational in nature, with some studies showing weak or inconsistent associations[26–28]. As a result, the current literature is not definitive and does not support inferences around causal direction. Moreover, there remains insufficient research investigating the association between simple functional habitual movements, such as walking pace, and LTL. Therefore, the aim of this study was to investigate the association between self-reported walking pace and LTL in middle-aged adults. This included harnessing previously defined genetic instruments for both walking pace and LTL to undertake bi-directional Mendelian randomisation (MR) analyses to help clarify the causal nature

and relative importance of any observed associations. To aid broader interpretation, we also supported observational analyses for self-reported walking pace using accelerometer-assessed total physical activity and intensity. Our observational hypothesis was that a brisker walking pace would be causally associated with longer LTL.

## Results

**Descriptive characteristics of the observational analytical sample**. Descriptive characteristics for the analysis sample are shown in Table 1. Mean age was 56.5 years (SD, 8.1); mean BMI was 27.2 kg/m$^2$ (SD, 4.65); and 54% and 95% were female and white, respectively. Approximately half the participants reported an average/steady walking pace ($n = 212,303$; 52.3%), with 6.6% ($n = 26,835$) reporting a slow walking pace and 41.1% ($n = 166,843$) reporting a brisk pace. Compared to slow walkers, those who reported being average/steady and brisk walkers were slightly younger, were more likely to have never smoked, and were less likely to be taking cholesterol/blood pressure medications, have a chronic disease, or have mobility limitations. Slow walkers reported engaging in less physical activity and had a higher deprivation index and prevalence of obesity compared to average and brisk walkers. Differences observed between the walking pace groups in the overall sample were mostly comparable for the accelerometer sub-sample (Supplementary Table 1).

**Observational associations of walking pace and accelerometer-derived physical activity with LTL**. The associations of walking pace with LTL are shown in Fig. 1. For the minimally-adjusted model (model 1), steady/average and brisk walkers had significantly longer LTL compared to slow walkers: standardised difference 0.066 (95% CI: 0.053–0.078) and 0.101 (0.088–0.113), respectively. Associations for steady/average and brisk walking pace remained statistically significant, but were attenuated, following adjustment for potential confounding variables: [model 2; steady average = 0.038 (0.025–0.051) and brisk = 0.058 (0.045–0.072)]. Further sequential adjustment for total self-reported physical activity volume (MET-min/week) and BMI (models 3 and 4) did not materially alter the results (Fig. 1).

**Accelerometer analyses**. Secondary analyses in the subset with accelerometer-derived continuous exposure measures of total physical activity and intensity ($n = 86,002$) are shown in Fig. 2. These show that undertaking a greater proportion of daily physical activity at a higher intensity (intensity gradient) was associated with longer LTL, with associations retained (albeit attenuated) after covariate adjustment. In contrast, there was less evidence for an association with total physical activity.

**Bi-directional MR analyses**. Table 2 shows the MR analysis for the association of LTL on walking pace (part 1) and the association of walking pace on LTL (part 2), both with and without adjustment for BMI within the walking pace GWAS. There was no evidence of a causal association of LTL on walking pace, with or without adjustment for BMI. However, there was evidence that walking pace is causally associated with LTL. Per modelled difference in walking pace category (slow to steady/average, or steady/average to brisk) there was an increase in LTL SD of 0.192 (95% CI: 0.077, 0.306) before and 0.226 (0.061–0.388) after adjustment for BMI. No significant evidence of directional pleiotropy was found using MR-Egger's intercept (Table 2). Other MR methods showed broadly consistent findings; sensitivity analyses using weighted median MR attenuated the results somewhat, but the MR robust adjusted profile score estimated stronger effects. As three variants in each genetic instrument

**Table 1 . Descriptive characteristics at baseline of the main analytical sample and by self-reported walking pace.**

| Characteristics | Total sample | Slow | Average/steady | Brisk |
|---|---|---|---|---|
| | N=405,981 | N=26,835 | N=212,303 | N=166,843 |
| Age (years), mean (SD) | 56.46 (8.10) | 58.95 (7.51) | 57.14 (8.03) | 55.20 (8.09) |
| Female gender, n (%) | 217,267 (53.5%) | 14,444 (53.8%) | 113,832 (53.6%) | 88,991 (53.3%) |
| White ethnicity, n (%) | 386,820 (95.3%) | 24,279 (90.5%) | 200,811 (94.6%) | 161,730 (96.9%) |
| Highest educational level achieved, n (%) | | | | |
| No qualification | 63,306 (15.6%) | 8,603 (32.1%) | 38,022 (17.9%) | 16,681 (10.0%) |
| Any other qualification | 203,534 (50.1%) | 12,518 (46.6%) | 109,862 (51.7%) | 81,154 (48.6%) |
| Degree level or above | 139,141 (34.3%) | 5,714 (21.3%) | 64,419 (30.3%) | 69,008 (41.4%) |
| Townsend indicator of multiple deprivation, median (IQR) | −2.23 (−3.69-0.33) | −0.83 (−3.01-2.49) | −2.21 (−3.67-0.37) | −2.40 (−3.79-0.08) |
| In employment, n (%) | 237,960 (58.6%) | 8,414 (31.4%) | 119,230 (56.2%) | 110,316 (66.1%) |
| Cigarette smoking, n (%) | | | | |
| Never | 223,350 (55.0%) | 11,876 (44.3%) | 114,422 (53.9%) | 97,052 (58.2%) |
| Previous | 141,890 (34.9%) | 10,416 (38.8%) | 75,768 (35.7%) | 55,706 (33.4%) |
| Current | 40,741 (10.0%) | 4,543 (16.9%) | 22,113 (10.4%) | 14,085 (8.4%) |
| Alcohol consumption, n (%) | | | | |
| Never or previous | 29,449 (7.3%) | 4,480 (16.7%) | 15,824 (7.5%) | 9,145 (5.5%) |
| < Twice a week | 193,748 (47.7%) | 14,238 (53.1%) | 105,542 (49.7%) | 73,968 (44.3%) |
| At least three times a week | 182,784 (45.0%) | 8,117 (30.2%) | 90,937 (42.8%) | 83,730 (50.2%) |
| Added salt intake, n (%) | | | | |
| Never/rarely | 228,030 (56.2%) | 12,960 (48.3%) | 114,910 (54.1%) | 100,160 (60.0%) |
| Sometimes or more frequent | 177,951 (43.8%) | 13,875 (51.7%) | 97,393 (45.9%) | 66,683 (40.0%) |
| Oily fish consumption, n (%) | | | | |
| More than once a week | 229,701 (56.6%) | 13,923 (51.9%) | 116,979 (55.1%) | 98,799 (59.2%) |
| Fruit and vegetable intake score, median (IQR) | 2.00 (1.00-2.00) | 1.00 (1.00-2.00) | 1.00 (1.00-2.00) | 2.00 (1.00-3.00) |
| Weekly frequency of red or processed meat intake, median (IQR) | 0.75 (0.50-1.25) | 0.88 (0.50-1.25) | 0.75 (0.50-1.25) | 0.75 (0.50-1.13) |
| Mean sleep duration, n (%) | | | | |
| <7 hours/day | 97,302 (24.0%) | 8,492 (31.6%) | 50,447 (23.8%) | 38,363 (23.0%) |
| 7-8 hours/day | 278,701 (68.6%) | 14,223 (53.0%) | 145,043 (68.3%) | 119,435 (71.6%) |
| >8 hours/day | 29,978 (7.4%) | 4,120 (15.4%) | 16,813 (7.9%) | 9,045 (5.4%) |
| Body mass index, n (%) | | | | |
| Normal weight (<25 kg/m2) | 137,660 (33.9%) | 4,214 (15.7%) | 57,872 (27.3%) | 75,574 (45.3%) |
| Overweight (25-30 kg/m2) | 174,368 (42.9%) | 8,854 (33.0%) | 95,206 (44.8%) | 70,308 (42.1%) |
| Obese (≥30 kg/m2) | 93,953 (23.1%) | 13,767 (51.3%) | 59,225 (27.9%) | 20,961 (12.6%) |
| Current prescription of blood pressure or cholesterol medicine, n (%) | 108,771 (26.8%) | 13,605 (50.7%) | 63,301 (29.8%) | 31,865 (19.1%) |
| Diagnosis of diabetes or insulin prescription, n (%) | 19,843 (4.9%) | 4,048 (15.1%) | 11,614 (5.5%) | 4,181 (2.5%) |
| Previous diagnosis of cardiovascular disease, n (%) | 26,214 (6.5%) | 5,400 (20.1%) | 14,566 (6.9%) | 6,248 (3.7%) |
| Previous diagnosis of cancer, n (%) | 33,965 (8.4%) | 3,207 (12.0%) | 18,077 (8.5%) | 12,681 (7.6%) |
| Mobility limitation, n (%) | 155,135 (38.2%) | 20,834 (77.6%) | 83,977 (39.6%) | 50,324 (30.2%) |
| Total MET-minutes/week physical activity, median (IQR) | 1668 (748-3372) | 813 (330-1980) | 1583 (702-3272) | 1944 (954-3715) |
| Total white blood cell (Leukocyte) count (10^9 cells/Litre), median (IQR) | 6.61 (5.61-7.80) | 7.30 (6.15-8.67) | 6.73 (5.71-7.91) | 6.40 (5.44-7.50) |
| Telomere length (z-score), mean (SD) | 0.004 (0.998) | −0.133 (1.016) | −0.022 (0.998) | 0.060 (0.992) |

Townsend score, a composite area-level measure of deprivation based on unemployment, non-car ownership, non-home ownership, and household overcrowding; a higher score.

showed some evidence of linkage disequilibrium ($r^2 > 0.1$) we removed these from each instrument as a sensitivity analysis, with the interpretation of findings unaffected (Supplementary Table 2).

## Discussion

In this large sample of middle-aged adults, we provide evidence that faster self-reported walking pace is associated with longer LTL. In support of the importance of walking pace, using accelerometer-assessed physical activity we showed that more time habitually spent in higher intensity activities (e.g. brisk walking) had a stronger association with LTL than total activity. Importantly, bi-directional MR analyses also suggest a causal link between walking pace and LTL, rather than the other way around. Overall, these findings support more intensive habitual movement, such as faster walking pace, as potentially important determinants of LTL and overall health status in humans.

The MR results provide important insights in the context of the current literature, which is limited by a lack of high-quality data and mixed findings from interventional research. The few studies that have been undertaken in this area are generally supportive of our findings. For example, in a non-randomised study, long-term endurance training was associated with reduced LTL erosion compared to healthy non-exercisers[29]. A randomised controlled trial in 68 caregivers (as a model of high exposure to stress) also found that 40 minutes of moderate-intensity aerobic exercise 3–5 times per week reduced LTL attrition compared to those in the control group[30]. Further, a recent meta-analysis of case control

and intervention studies suggested that, across 21 included studies, the effect of exercise training produced a moderate effect size (0.7), with exercise associated with longer telomeres[28]; however, associations were no longer statistically significant after accounting for publication bias and stratification/subgrouping. These findings are also supported by mechanistic studies in animal models showing that chronic exercising induced increases in telomere binding proteins (shelterin) and telomerase enzyme activity[31–33]. This acts to protect telomere degradation, with an associated reduction of apoptosis and cell-cycle arrest in the myocardium, whilst also attenuating age-related erosion of telomeres in hepatocytes and cardiomyocytes[29,31–33]. Other hypothesised mechanisms linking habitual physical activity with telomere protection include potential changes in oxidative stress, inflammation, and decreased skeletal muscle satellite cell content[31,32,34].

It is noteworthy that associations were observed for walking pace, independent of physical activity volume, with accelerometer data also indicating associations for physical activity intensity, but not for physical activity volume. Given that walking pace is a measure of physical function and closely associated with cardiorespiratory fitness[8], these results are consistent with findings for cardiovascular disease and all-cause mortality more generally, where measures of cardiorespiratory fitness have been shown to be stronger predictors of outcomes than measures of physical activity alone[8,35–37]. Furthermore, cardiorespiratory fitness is considered a cardiovascular vital sign with low fitness and frailty closely linked to aging[37,38]. Therefore, it is possible that the mechanisms linking physical activity to biological aging detailed

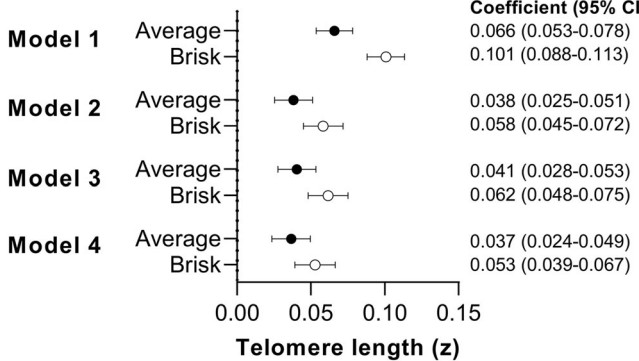

**Fig. 1 Associations between self-reported walking pace and LTL.** Data presented as β-coefficient (95% CI) for "average" (n = 212,032) and "brisk" (n = 166,641) walking pace relative to "slow" (26,804) walking pace (reference). Model 1: adjusted for age, sex, ethnicity, white blood cell count. Model 2: model 1 additionally adjusted for education level, employment status, Townsend index of deprivation, fruit and vegetable intake, processed and red meat intake, oily fish intake, regularity of adding salt to food, alcohol intake, smoking status, average sleep duration, blood pressure or cholesterol medication use, diabetes diagnosis or insulin prescription, mobility limitation, and prevalent cardiovascular disease and prevalent cancer. Model 3: model 2 additionally adjusted for total physical activity volume (MET-min/week). Model 4: model 3 additionally adjusted for body mass index.

above may be optimised by exercise interventions designed specifically to increase cardiorespiratory fitness and physical function, rather than overall physical activity volume per se. However, this hypothesis needs further investigation.

Interestingly, the suggested causal effect of walking pace on LTL via MR was greater than that suggested by the adjusted observational associations reported in this study, or for exercise intervention studies in middle aged adults[28]. We have previously shown that each additional year in chronological age is associated with a Z-standardised LTL value of −0.024[39]. Therefore, the difference in LTL between slow and fast walkers suggested by the MR analysis is equivalent to 16 years of age-related difference in LTL, whereas the adjusted observational analysis produced estimates that were equivalent to 2 years of age-related difference. There are several possible explanations for difference. Previously investigated associations between other risk factors and health outcomes, such as blood pressure and lipids, have also reported greater effect sizes from MR than those proposed by observational or interventional research[40–42]. It has been proposed that this difference could be explained by MR measuring lifetime exposure, whereas observational studies measure the exposure at a single point in time, and/or intervention research assesses change over a relatively short period of time[43]. However, it has also been cautioned that the association of genetic variants with risk factors may vary by age which could act to inflate MR estimates[44]. Furthermore, differences in the strength of the genetic association with the exposure and outcome may also influence MR estimates[44]. Therefore, whilst MR can help determine causality, the size of the effect should be interpreted with caution and is likely to be greater than the magnitude of change that can be anticipated from any future intervention.

Key strengths of this analysis are the large, contemporary, well-phenotyped cohort with high quality LTL data, and the use of bidirectional MR to examine potential and relative causal effects. However, there are some important limitations. Our study is based on a one-time measurement of LTL, with the inherent limitations of cross-sectional analysis. Longitudinal studies are necessary to assess any relationship between physical activity and LTL attrition. However, it is notable that longitudinal studies of LTL in adults

suggest very modest changes within an individual over time, particularly over shorter rather than whole life periods[45]. The importance of lifetime exposure was further supported by the difference in association between the observational and MR analyses. In addition, whilst the self-reported measure of walking pace in UK Biobank has been associated with objectively measured cardiorespiratory fitness[8], it is possible that responses are also influenced by wider factors and personality traits that will not be affected by an intervention to target walking pace per se. Nevertheless, analysis of accelerometer data in UK Biobank supported the findings for walking pace, as a measure of habitual movement intensity was associated with longer LTL, whereas there was little evidence of an association for overall movement volume. Although our observational findings were supported by MR, this should be viewed as suggestive of causality, rather than confirmative. The effect sizes and strength of association were attenuated when using the MR-median weighted method, perhaps indicating evidence of potential bias in the walking pace instrument. However, this may be due to reduced power in the method, as our other sensitivity analysis using MR-robust adjusted profile scores was still highly significant. Finally, although large in scale, the UK Biobank cohort is healthier than the general population, and the accelerometer sub-study may be subject to some further selection biases (i.e., being measured a median of 5.7 years after UKBB baseline assessment). However, key covariates have been shown to be mostly stable over this time period[46], and risk factor associations have previously been shown to be generalisable to the general population[47].

A faster habitual walking pace may be causally associated with longer LTL and could help explain some of the beneficial effects of brisk walking on health status. Further research should confirm whether behavioural interventions focused on increasing walking pace or physical activity intensity could act to slow the erosion of LTL. Future work should also elucidate whether these findings simply add support to the use of self-reported walking pace as a measure of overall health status, with a slow walking pace identifying those with potentially accelerated biological ageing, and thus a priority group for other lifestyle/pharmaceutical interventions. Characterising the nature of associations between walking pace and LTL in different population sub-groups, particularly those at increased risk of chronic disease or unhealthy ageing, will also be important.

## Methods

**Data source and study population**. This analysis used data from participants within UK Biobank, a large prospective cohort of middle-aged adults designed to support biomedical analysis focused on improving the prevention, diagnosis, and treatment of chronic disease through phenotyping and genomics data[48]. Between March 2006 and July 2010, individuals living within 25 miles of one of the 22 study assessment centres located throughout England, Scotland, and Wales were recruited and provided comprehensive data on a broad range of demographic, clinical, lifestyle, and social outcomes. All participants provided written informed consent and the study was approved by the NHS National Research Ethics Service (Ref: 11/NW/0382). Details of recruitment and measurements used to obtain data for this resource can be found on the UK Biobank website: https://www.ukbiobank.ac.uk.

**Ethics approval and consent to participate**. The UK Biobank study received ethical approval from the North West England Research Ethics Committee (Ref: 16/NW/0274). Participants gave informed consent before participation.

**Self-reported walking pace**. Self-reported walking pace was ascertained using a touchscreen question: "How would you describe your usual walking pace?" with response options of "slow", "steady/average" or "brisk". Participants could access further information which defined a slow pace as less than 3 miles per hour, a steady/average pace as between 3–4 miles per hour, and a brisk pace as more than 4 miles per hour." We excluded participants whose answers were "None of the above" or "Prefer not to answer" (n = 3956).

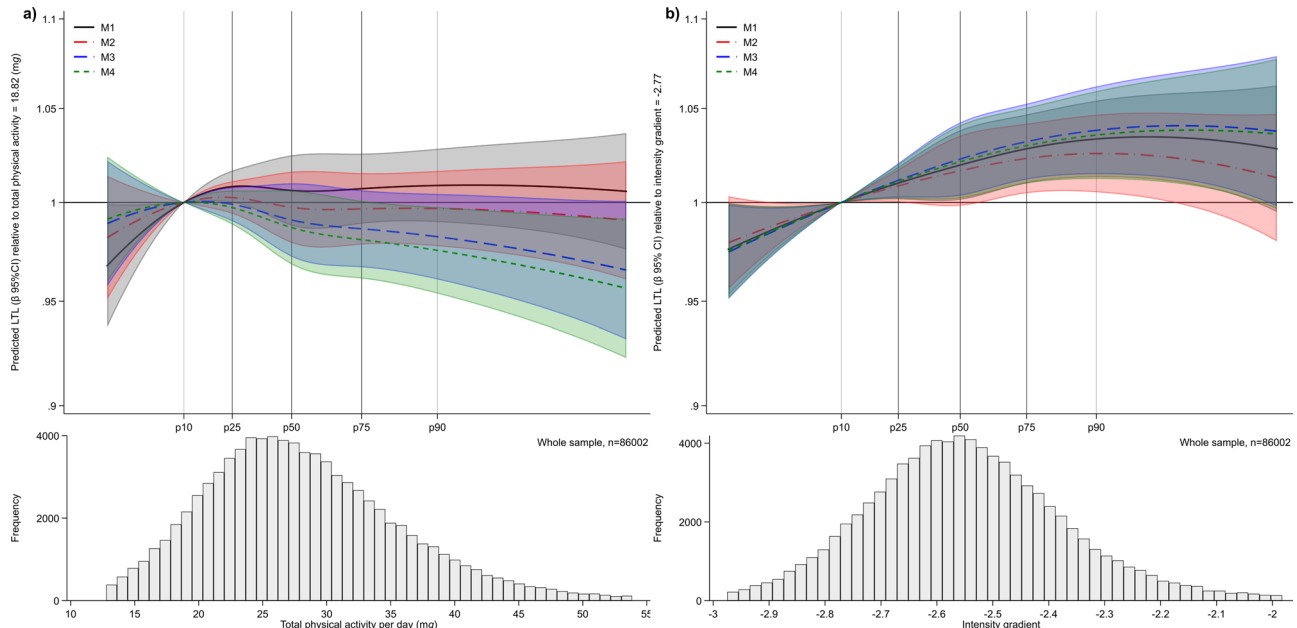

**Fig. 2 Baseline exposure distribution and predicted β-coefficient (95% CI), relative to the 10th percentile reference of each exposure.** Data presented for the association of **a** total physical activity and **b** intensity gradient with LTL ($n = 86,002$). Models were fitted with the use of restricted cubic splines (3 evenly-spaced knots). Predicted LTL β-coefficients and histogram data shown for values between the 1st or 99th percentiles of each exposure distribution. The reference values chosen for the exposure are at the 10th percentile (total physical activity = 18.82 m*g*; intensity gradient −2.77), with 25th, 50th, 75th and 90th percentiles also denoted. A higher (less negative) intensity gradient (intensity distribution of physical activity) indicates more time is habitually spent in higher intensity activities (e.g., brisk walking) over a day. Model 1: is adjusted for age, sex, ethnicity, white blood cell count. Model 2: model 1 additionally adjusted for education level, employment status, Townsend index of deprivation, season of accelerometer wear, fruit and vegetable intake, processed and red meat intake, oily fish intake, regularity of adding salt to food, alcohol intake, smoking status, average sleep duration, blood pressure or cholesterol medication use, diabetes diagnosis or insulin prescription, mobility limitation, and prevalent cardiovascular disease and prevalent cancer. Model 3: model 2 additionally adjusted for either intensity gradient or total physical activity (i.e. mutual adjustment). Model 4: model 3 additionally adjusted for body mass index.

---

**Table 2 Mendelian randomisation between self-reported walking pace and LTL.**

**Part 1: LTL on walking pace (121 SNPs for LTL)**

|  | Unadjusted walking pace I² = 61.8% | | | Walking pace adjusted for BMI I² = 41.8% | | |
|---|---|---|---|---|---|---|
|  | Beta* (95% CI) | *P*-value | Egger - p | Beta* (95% CI) | *P*-value | Egger - p |
| MR-IVW | 0.007 (−0.009, 0.023) | 0.383 | 0.965 | 0.006 (−0.006, 0.018) | 0.316 | 0.978 |
| MR-WM | 0.003 (−0.013, 0.020) | 0.687 |  | 0.001 (−0.014, 0.017) | 0.852 |  |
| MR-RAPS | 0.007 (−0.003, 0.017) | 0.151 |  | 0.006 (−0.003, 0.016) | 0.185 |  |

**Part 2: Walking pace on LTL (70 SNPs for walking pace)**

|  | Unadjusted walking pace I² = 72.9% | | | Walking pace adjusted for BMI I² = 74.1% | | |
|---|---|---|---|---|---|---|
|  | Beta† (95% CI) | *P*-value | Egger - p | Beta† (95% CI) | *P*-value | Egger - p |
| MR-IVW | 0.192 (0.077, 0.306) | 0.001 | 0.563 | 0.226 (0.063, 0.388) | 0.006 | 0.157 |
| MR-WM | 0.112 (0.016, 0.209) | 0.023 |  | 0.110 (−0.021, 0.242) | 0.101 |  |
| MR-RAPS | 0.211 (0.151, 0.271) | 4.44E-12 |  | 0.270 (0.187, 0.353) | 1.51E-10 |  |

Where MR-IVW is the inverse-variance weighted MR which was used as the primary MR method, with MR-WM as the weighted-median MR and MR-RAPS as the robust adjusted profile score MR which were both included as sensitivity analyses. Beta* is the estimated unit difference in walking pace per 1 SD increase in LTL, and Beta† is the SD change in LTL per 1 unit difference in walking pace, where a 1 unit increase in self-rated walking pace represents a change in category from slow to steady/average, or from steady/average to brisk pace.

---

**Covariate measurement**. We utilised demographic and lifestyle related characteristics of age, sex, ethnicity (white/non-white), Townsend Index of deprivation, highest educational level achieved (degree or above/any other qualification/no qualification), employment status (unemployed/in paid or self-employment), alcohol drinking status (never/previous/current), salt added to food (never/sometimes), oily fish intake (never/sometimes), fruit and vegetable intake (a score from 0–4 taking into account questions on cooked and raw vegetables, fresh and dried fruit consumption), processed and red meat intake (average weekly frequency in days per week), and sleep duration (<7, 7–8, >8 hours), and a diagnosis of cardiovascular disease or cancer prior to baseline. The latter two prevalent disease variables were derived from the self-reported history of heart attack, angina, stroke, or cancer variables, and from linked hospital episode data (corresponding ICD 10 codes I20-25, I60-69, or C00-99). Health-related covariates of blood pressure and cholesterol

medication, doctor diagnosed diabetes or prescribed insulin medication and mobility limitations (self-reported longstanding illness or disability, chest pain at rest, or leg pain while walking), white blood cell (leucocyte) count, and body mass index (BMI) in three categories (<25, 25-30, ≥30 kg/m²) were included in models. Total MET-minutes/week of physical activity was derived from weekly frequency of walking, moderate, or vigorous intensity physical activity using the short-form International Physical Activity Questionnaire[49]. Further details for each variable are available on the UK Biobank Website https://www.ukbiobank.ac.uk/

**Accelerometer physical activity measurements**. Physical activity was assessed using accelerometry in a sub-sample of ~100,000 adults between June 2013 and December 2015. Participants were invited to wear an Axivity AX3 tri-axial

accelerometer (Axivity, Newcastle, UK) 24 hours a day on their dominant wrist for seven consecutive days[50]. For each participant, we extracted the accelerometer data (5-second epoch time series) from UK Biobank[50] and converted it to R-format for processing and analysis with GGIR (version 1.11-0, http://cran.r-project.org)[51]. Participants were excluded if they failed calibration (including those not calibrated on their own data), had fewer than three days of valid wear (defined as >16 h per day), or wear data were not present for each 15 minute period of the 24 h cycle. Data from $n = 86,002$ UK Biobank participants with valid accelerometer and LTL data, and complete covariate data were included (see Supplementary Fig. 1).

Accelerometer measures, selected to describe total physical activity and its intensity, were average acceleration over the 24 h day (proxy for total physical activity, m$g$) and intensity gradient over 24 h (a measure of the intensity distribution of physical activity over the day)[50–52]. A higher average acceleration indicates more physical activity is accumulated across the day, irrespective of the intensity. A higher intensity gradient indicates more time is habitually spent in higher intensity activities, such as brisk walking. The intensity gradient describes the negative curvilinear relationship between physical activity intensity and the time accumulated at that intensity (Supplementary Fig. 2). It is always negative, reflecting the decrease in time accumulated as intensity increases. Findings for the intensity distribution of physical activity derived from the accelerometer data can be interpreted as being supportive of walking pace, as both relate to habitual movement intensity.

**Leucocyte telomere length measurements.** LTL was measured using an established multiplex qPCR assay from 488,415 available DNA samples of participants in UK Biobank, which are detailed elsewhere[39]. After extensive quality checks and adjustment for technical factors, valid LTL measurements were available for 472,577 individuals[39]. For analyses of data available for the full cohort, we used log-transformed and z-standardised LTL values (UK Biobank data field code 22192), of which log-transformed data were re-standardised for analyses performed on the sub-set of participants with accelerometer data.

**Statistics and Reproducibility.** For analyses presented in this paper, we included all participants with LTL measured from the UK Biobank baseline sample, where there was no mismatch in self-reported and genetic sex ($n = 472,248$). Exclusions were also made for missing walking pace or covariate data, or for missing accelerometer data in the subset analysis (Supplementary Fig. 1).

A series of linear regression models were used to quantify the associations of "steady/average" and "brisk" self-reported walking pace with Z-standardised log-LTL ($\beta$-coefficient with 95% CI), compared to "slow" walkers as the reference group. Model 1 adjusted for age, sex, ethnicity and total white blood cell count, as these variables are known to be associated with LTL[39]. Model 2 additionally adjusted for other included confounders. Models 3 and 4 additionally adjusted for total physical activity and then body mass index, which were considered last given their potential role as confounders or mediators.

Bi-directional one-sample Mendelian randomisation (MR) analysis (see Supplementary Fig. 3) were conducted to evaluate a potential causal relationship between longer LTL and self-rated walking pace, using the inverse-variance weighted[53] method with sensitivity analyses applying both the weighted median[54] and robust adjusted profile score[55] methods. We also used MR-Egger regression to assess robustness to horizontal pleiotropy[56]. Each of these approaches makes a slightly different set of assumptions about the pleiotropic effects of genetic instruments, hence if the effect estimates are consistent across methods this provides stronger evidence of causality. To examine the causal association of LTL on walking pace (part 1), the LTL instrument utilised a set of 130 genome-wide significant ($P < 8.31 \times 10^{-9}$), conditionally independent, uncorrelated, and non-pleiotropic genetic variants we recently identified as genetic instruments for LTL[18]. We matched variants to the publicly available walking pace GWAS data that were unadjusted and adjusted for BMI, matching 121 variants. For walking pace on LTL (part 2), we considered 70 genome-wide significant ($P < 5.0 \times 10^{-8}$) independent genetic variants as the instrument from the unadjusted GWAS using weights from both the unadjusted GWAS and the BMI-adjusted GWAS[15]. These were matched to the LTL GWAS[18], matching all variants. To interpret the causal effect estimate of 1-SD increased LTL on differences in walking pace category, the coded values 0, 1 and 2 for self-reported slow, steady/average and brisk walking pace can be thought of as threshold values for an underlying continuous trait, as has been demonstrated previously[15].

To support the findings and interpretation for self-reported walking pace, we included secondary analyses examining the sub-set of the UK Biobank cohort with accelerometer data and LTL ($n = 86,002$; see Supplementary Fig. 1 for flowchart and Supplementary Table 1 for sub-sample descriptive characteristics), focussing on two key metrics summarising total physical activity (average acceleration) and the intensity distribution (intensity gradient) of physical activity over each 24-hour day. Due to some evidence of non-linearity, associations for these two accelerometer exposures with LTL were examined using restricted cubic splines (three evenly-spaced knots), with reference values set at the 10th percentile of the exposure.

Observational analyses were conducted using Stata v15.1 (StataCorp, TX, USA) and statistical significance was set at $p < 0.05$ (two-tailed). MR analyses were performed using the *MendelianRandomisation* package implemented in R software[57].

**Reporting summary**. Further information on research design is available in the Nature Research Reporting Summary linked to this article.

## Data availability
The UK Biobank data that support the findings of this study can be accessed by researchers on application (https://www.ukbiobank.ac.uk/register-apply/). Individual-level genotype data are available by application to the UK Biobank. Variables derived specifically for this study will be returned along with the code to the UK Biobank for future applicants to request. No additional data are available.

## Code availability
Analysis code is available on request to the corresponding author.

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

## Acknowledgements

We are grateful to the participants of the UK Biobank Study and those who collected and manage the data. This research has been conducted using the UK Biobank Resource under Application #33266 and LTL analysis was funded by the UK Medical Research Council (MRC), Biotechnology and Biological Sciences Research Council and British Heart Foundation (BHF) through MRC grant MR/M012816/1. Accelerometer data processing was supported by the Lifestyle Theme of the Leicester NHR Leicester Biomedical Research Centre and NIHR Applied Research Collaborations East Midlands (ARC-EM). CPN is funded by the British Heart Foundation.

## Author contributions

P.C.D., A.V.R., V.C., C.P.N., T.Y. and N.J.S. developed the research question. V.C. supervised the LTL analysis. P.C.D. undertook the observational analyses and drafted the manuscript. C.P.N., P.C.D., and C.M. undertook the MR analyses. All authors, including M.D., K.K., C.R., I.T., and F.Z., contributed to the interpretation and revised the manuscript for important intellectual content.

## Competing interests

The authors declare no competing interests.
