## [Peer Review File · Communications Biology]

Reviewers' comments:

Reviewer #1 (Remarks to the Author):

I thank the authors for presenting this important study. You can find my comments on the study below.

Bartu Eren Güneşliol

This study, which includes biobank data, seems to be valuable. There are studies on extensive population data on physical activity and telomere length. Studies based on the National Health and Nutrition Examination Survey (NHANES) data are exemplary in this regard. This study includes self-report data of individuals, as mentioned in the article as a limitation by authors. However, this study may offer a new and different perspective to the literature. First, this study has also Accelerometer-assessed measures of total physical activity and intensity, as well as self-reported walking pace data. Accelerometer data is valuable. This makes the study even more critical. Second, the study also put forward a causal association of walking pace on LTL. Therefore, with these aspects, the study would contribute to the limited literature on this subject. The statistical analysis of the study and the technical issues they followed in terms of study design seemed appropriate to me. I find it useful to accept the manuscript by considering the minor revisions I have mentioned below (also in the reviewers' attachment file):

Page 3, line 86: Authors should write align with instead of align to.

Page 3, line 109: This part of that sentence 'The aim of this study was, therefore,' should be better written as 'Therefore, this study aimed...'

Page 3, line 110: middle-aged

Page 3, lines 113: We also supported...

Page 4, line 114: Remove the comma to fix punctuation at the end of the word '...intensity,'

Page 5, line 171: Add a comma after the word 'factors' to fix punctuation.

Page 5, line 172: Add a comma after the word 'cohort' to fix punctuation.

Page 5, line 199: Add a comma after the word 'part 1' to fix punctuation.

Page 5, line 204: Add a comma after the word 'part 2' to fix punctuation.

Page 5, line 207: Delete the word 'length' as LTL already means leukocyte telomere length.

Page 6, line 242: Add a comma after the word 'model 1' to fix punctuation.

Page 6, line 255: Add a comma after the word 'contrast' to fix punctuation.

Page 7, line 288: Add a comma after the word 'studies' to fix punctuation.

Page 7, line 292: The authors did not mention 'telomerase' till here. This may confuse readers' minds. So, it should be better that authors explain this word briefly here.

Page 7, lines 293-295: Is that sentence explain the increased expression of telomerase and shelterin complex? It looks like it is. However, readers may not understand it. So, this part can be written more clearly.

Page 8, line 311: relatively

Page 8, lines 346-348: In this part, the authors may also think to slightly emphasise age and race in the concept of different populations.

Reviewer #2 (Remarks to the Author):

The work by Dempsey et al. uses Mendelian Randomization (MR) to examine bidirectional causal relationship between self-reported walking and leukocyte telomere length (LTL) using data from the UK Biobank study. The authors found that self-reported steady/average and brisk walking was associated with longer LTL compared to slow walking. In addition, using accelerometer data, authors found that intensity, but not total physical activity, was associated with longer LTL. At last, MR analysis showed that genetically determined walking pace was associated with longer LTL.

The results add important evidence to support a causal link between walking and LTL. However, there are several considerations to be addressed:

1. Is there any overlap between genetic instruments of LTL and walking pace?
2. Did the authors do any sensitivity analysis to evaluate potential violations of MR assumptions?
3. How much of heritability of walking pace and LTL is explained by their respective genetic instruments? I wonder if lack of significant results of the LTL genetic instrument and walking pace is in part due to low heritability prediction.
4. A major limitation of the study is the one-time measure of LTL. Evidence suggest that LTL attrition rate may be more relevant for aging process. It would be relevant for readers if authors discuss this limitation.

Reviewer #3 (Remarks to the Author):

Dempsey et al., describe an interesting study in which investigators explore the correlation between leucocyte telomere length (LTL) with measures of duration and intensity of physical activity. Prior study (Codd et al., Nature Genetics Oct 2021) have described associations with physiological traits and diseases in relations to LTL; this study appears to complement prior studies in describing LTL to a functional parameter (walking pace) of health that gives unique insight into their mobility and overall frailty.

1. Authors conclude that the speed of walking pace is causally related with longer LTL (page 8, Lines 340-348) but what would the mechanistic explanation of the pace impacting LTL rather than the total duration of activity? The authors should discuss the implications of their findings more in depth
2. Table 1: Subjects are described based on the frequency of alcohol and cigarette consumption- would be more meaningful if they have access to the amount of alcohol or the number of cigarettes used as it would reflect cumulative nature of environmental insults on telomere length
3. Study limitations include: (1) heavily Caucasian/white population (95%) with restricted generalizability to other ethnic populations (2) selection bias given limited analysis within UK Biobank participants- limited information is provided on recruitment process for those readers who are not familiar with the UK biobank.
4. Authors should describe and give more information on what constitutes as "mobility limitation" described among their subjects (39% of their total sample, table 1) and comment on whether these limitations impact their observations and study conclusion

Wednesday, 2 February 2022

MS ID#: COMMSBIO-21-2571-T

Causal Associations of Self-Reported Walking Pace with Telomere Length in 405,981 middle-aged adults: a UK Biobank study

Dear Dr. H el ene Choquet,

We greatly appreciate the opportunity to revise and resubmit our paper to *Communications Biology*. We found the detailed comments and feedback thoughtful and constructive and believe that we have been able to address these and significantly strengthen the manuscript. Below, we delineate the responses and changes point by point, which address each Reviewer's comments.

Our responses are provided in **blue text** below each comment. We have numbered each response with the initial Arabic numeral indicating the reviewer number: e.g., 1.1, 2.3, etc. Text contained within the boxes is used to identify the specific parts in the manuscript that have been revised or added (**modified text is highlighted in yellow** or **tracked changes** where appropriate). These revisions are also highlighted using track changes within the main manuscript documents.

We hope that the present format will help to make subsequent aspects of the assessment process more streamlined.

Thank you again for considering our work.

Yours sincerely

Paddy Dempsey (on behalf of all authors)

- Medical Research Council Epidemiology Unit

University of Cambridge School of Clinical Medicine, Box 285 Institute of Metabolic Science, Cambridge Biomedical Campus, Cambridge CB2 0QQ

- Email: Paddy.Dempsey@mrc-epid.cam.ac.uk

REVIEWER 1

General comments:

I thank the authors for presenting this important study. You can find my comments on the study below.

This study, which includes biobank data, seems to be valuable. There are studies on extensive population data on physical activity and telomere length. Studies based on the National Health and Nutrition Examination Survey (NHANES) data are exemplary in this regard. This study includes self-report data of individuals, as mentioned in the article as a limitation by authors. However, this study may offer a new and different perspective to the literature. First, this study has also Accelerometer-assessed measures of total physical activity and intensity, as well as self-reported walking pace data. Accelerometer data is valuable. This makes the study even more critical. Second, the study also put forward a causal association of walking pace on LTL. Therefore, with these aspects, the study would contribute to the limited literature on this subject. The statistical analysis of the study and the technical issues they followed in terms of study design seemed appropriate to me. I find it useful to accept the manuscript by considering the minor revisions I have mentioned below (also in the reviewers' attachment file):

Thank you for your positive appraisal of our paper and for the detailed comments.

Comment #1.1

Page 3, line 86: Authors should write align with instead of align to.

Page 3, line 109: This part of that sentence 'The aim of this study was, therefore,' should be better written as 'Therefore, this study aimed...'

Page 3, line 110: middle-aged

Page 3, lines 113: We also supported...

Page 4, line 114: Remove the comma to fix punctuation at the end of the word '...intensity,'

Page 5, line 171: Add a comma after the word 'factors' to fix punctuation.

Page 5, line 172: Add a comma after the word 'cohort' to fix punctuation.

Page 5, line 199: Add a comma after the word 'part 1' to fix punctuation.

Page 5, line 204: Add a comma after the word 'part 2' to fix punctuation.

Page 5, line 207: Delete the word 'length' as LTL already means leukocyte telomere length.

Page 6, line 242: Add a comma after the word 'model 1' to fix punctuation.

Page 6, line 255: Add a comma after the word 'contrast' to fix punctuation.

Page 7, line 288: Add a comma after the word 'studies' to fix punctuation.

These edits have all been addressed.

Comment #1.2

Page 7, line 292: The authors did not mention 'telomerase' till here. This may confuse readers' minds. So, it should be better that authors explain this word briefly here.

Page 7, lines 293-295: Is that sentence explain the increased expression of telomerase and shelterin complex? It looks like it is. However, readers may not understand it. So, this part can be written more clearly.

We have updated this text to clarify/simplify.

Comment #1.3

Page 8, line 311: relatively

Page 8, lines 346-348: In this part, the authors may also think to slightly emphasise age and race in the concept of different populations.

Thank you. These edits have been addressed. There are many different sub-groups that could also be examined (e.g., including age and ethnicity), but we chose to stick with our original emphasis to additional sub-populations mentioned (i.e., chronic disease, unhealthy ageing).

REVIEWER 2

General comments:

The work by Dempsey et al. uses Mendelian Randomization (MR) to examine bidirectional causal relationship between self-reported walking and leukocyte telomere length (LTL) using data from the UK Biobank study. The authors found that self-reported steady/average and brisk walking was associated with longer LTL compared to slow walking. In addition, using accelerometer data, authors found that intensity, but not total physical activity, was associated with longer LTL. At last, MR analysis showed that genetically determined walking pace was associated with longer LTL.

The results add important evidence to support a causal link between walking and LTL. However, there are several considerations to be addressed:

Comment #2.1

Is there any overlap between genetic instruments of LTL and walking pace?

We thank the reviewer for this query. Although we had removed pleiotropic variants from the LTL instrument, on further inspection we found three variants that represent overlapped signals between the instruments ($r^2 > 0.1$). To test any potential impact of this overlap we removed these variants from both instruments and performed the MR again. As expected, the results were largely unaltered by the removal of 3 variants, with a very modest increase in significance in the effect of walking pace on LTL (full MR beta = 0.192, $p = 0.001$; 3 SNPs removed beta = 0.186, $p = 1.99 \times 10^{-4}$). We have now acknowledged the overlap within the manuscript and have added a sentence to the results section highlighting this (page 7, para 1):

Results, page 7: "... No significant evidence of directional pleiotropy was found using MR-Eggers intercept (Table 2). Other MR methods showed broadly consistent findings; sensitivity analyses using weighted median MR attenuated the results somewhat, but the MR robust adjusted profile score estimated stronger effects. **As three variants in each genetic instrument**

showed some evidence of linkage disequilibrium ($r^2 > 0.1$) we removed these from each instrument as a sensitivity analysis, with the interpretation of findings unaffected (Supplemental Table S2).”

Comment #2.2

Did the authors do any sensitivity analysis to evaluate potential violations of MR assumptions?

As stated in the manuscript (page 7, para 1) we performed the MR-IVW method as the primary analysis, but also conducted sensitivity analyses using MR-WM, MR-RAPS and MR-Egger. These results are also detailed in Table 2. We saw consistency in results across all methods, with no evidence of pleiotropy.

Comment #2.3

How much of heritability of walking pace and LTL is explained by their respective genetic instruments? I wonder if lack of significant results of the LTL genetic instrument and walking pace is in part due to low heritability prediction.

The Reviewer raises an interesting point. The SNP based heritability of walking pace is 13.6% and for LTL is 8.1%. Whilst the heritability of walking pace is slightly higher, we do not feel that this level of difference would explain the lack of an effect of LTL on walking pace. Indeed, if walking pace is adjusted for BMI the estimated heritability is reduced to 8.9%, comparable to that of LTL, yet the effect of walking pace on LTL remains significant, whilst that of LTL on walking pace remains null.

Comment #2.4

A major limitation of the study is the one-time measure of LTL. Evidence suggest that LTL attrition rate may be more relevant for aging process. It would be relevant for readers if authors discuss this limitation.

Thank you for this comment. We have added additional text to the manuscript to address this potential issue (see below). As mentioned, considering the size of effect seen in the MR compared to that of the observational analyses, we believe that this may reflect the effects of walking pace on LTL that have accrued over the lifetime, as we discuss on page 7, para 4. Studies have also suggested that changes within an individual are incredibly small, even over long time periods, for example 10 years. This challenges the suggestion that LTL attrition drives aging processes and suggests that LTL set in early life is more influential in driving age-related disease.

Discussion, page 8: “... Key strengths of this analysis are the large, contemporary, well-phenotyped cohort with high quality LTL data, and the use of bidirectional MR to examine potential and relative causal effects. However, there are some important limitations. Our study is based on a one-time measurement of LTL, with the inherent limitations of cross-sectional analysis. Longitudinal studies are necessary to assess any relationship between PA and LTL attrition. However, it is notable that longitudinal studies of LTL in adults suggest very modest changes within an individual over time, particularly over shorter rather than whole life periods [51]. The importance of lifetime exposure was further supported by the difference in association between the observational and MR analyses. In addition, whilst the self-reported measure of walking pace in UK Biobank...”

Comment #2.2

REVIEWER 3

General comments:

Dempsey et al., describe an interesting study in which investigators explore the correlation between leucocyte telomere length (LTL) with measures of duration and intensity of physical activity. Prior study (Codd et al., Nature Genetics Oct 2021) have described associations with physiological traits and diseases in relations to LTL; this study appears to complement prior studies in describing LTL to a functional parameter (walking pace) of health that gives unique insight into their mobility and overall frailty.

Comment #3.1

Authors conclude that the speed of walking pace is causally related with longer LTL (page 8, Lines 340-348) but what would the mechanistic explanation of the pace impacting LTL rather than the total duration of activity? The authors should discuss the implications of their findings more in depth

Thank you for your comments and for this suggestion. In the second paragraph of the Discussion, we comment on some of the potential or (previously) hypothesised mechanisms that may underly the associations between PA and LTL (e.g. telomerase activity, oxidative stress, inflammation, and decreased skeletal muscle satellite cell content). It was interesting to see that the accelerometer results further supported a role for PA intensity, independent of PA volume. However, it is challenging to further speculate on the mechanisms underlying why faster walking pace and/or PA intensity may distinctly impact on LTL, or whether there are specific thresholds beyond which PA intensity is no longer beneficial. We have now added an additional section to the Discussion highlighting the context and implications for this finding, with additional text around the potential importance of fitness and expansion of the mechanisms previously discussed.

Discussion, page 7: "...It is noteworthy that associations were observed for walking pace, independent of PA volume, with accelerometer data confirming associations for PA intensity, but not for PA volume. Given that walking pace is a measure of physical function and closely associated with cardiorespiratory fitness [8], these results are consistent with findings for cardiovascular disease and all-cause mortality more generally, where measures of cardiorespiratory fitness have been shown to be stronger predictors of outcomes than measures of PA alone [8, 46-48]. Furthermore, cardiorespiratory fitness is considered a cardiovascular vital sign with low fitness and frailty closely linked to aging [48, 49]. Therefore, it is possible that the mechanisms linking PA to biological aging detailed above may be optimised by exercise interventions designed specifically to increase cardiorespiratory fitness and physical function, rather than overall PA volume per se. However, this hypothesis needs further investigation...."

Comment #3.2

Table 1: Subjects are described based on the frequency of alcohol and cigarette consumption- would be more meaningful if they have access to the amount of alcohol or the number of cigarettes used as it would reflect cumulative nature of environmental insults on telomere length

Thank you for this comment. We are somewhat limited by the nature of the variables reported in UK Biobank. Alcohol variables that cover the main UK Biobank sample are derived based

on questions about the frequency of intake, which is reflected in our analysis. Additional manipulation and assumption would be required to turn this into units data. For the Reviewers interest, the variables available in the main cohort and subsets can be accessed through the showcase browser <https://biobank.ndph.ox.ac.uk/showcase/>. For smoking, some derivative work has been done on pack-years, but we do not have immediate access to these variables within our working dataset. Either way, we do not expect that it would have any material impact on the observational associations. It is worth highlighting that as with all observational data, there is a risk of residual confounding, particularly with self-reported data. This risk would remain with more (or differently detailed) measures of alcohol and smoking behaviour. Therefore, we aimed to draw conclusions from our work not from the observational study alone, but in consideration of the findings from the Mendelian Randomisation analyses, which is not subject to the same constraints around residual confounding.

Comment #3.3

Study limitations include: (1) heavily Caucasian/white population (95%) with restricted generalizability to other ethnic populations (2) selection bias given limited analysis within UK Biobank participants- limited information is provided on recruitment process for those readers who are not familiar with the UK biobank.

We have now provided a few further details to the Methods section for the reader (see below); however, most of this information is easily accessible online.

Methods, page 4: "... Between March 2006 and July 2010, individuals living within 25 miles of one of the 22 study assessment centres located throughout England, Scotland, and Wales were recruited and provided comprehensive data on a broad range of demographic, clinical, lifestyle, and social outcomes..."

.... "All participants provided written informed consent and the study was approved by the NHS National Research Ethics Service (Ref: 11/NW/0382). Details of recruitment and measurements used to obtain data for this resource can be found on the UK Biobank website: <https://www.ukbiobank.ac.uk...>"

Comment #3.4

Authors should describe and give more information on what constitutes as "mobility limitation" described among their subjects (39% of their total sample, table 1) and comment on whether these limitations impact their observations and study conclusion

The mobility limitation statistics reported in Table 1 are described in more detail in the Methods (covariates measurement) section on page 4. These include "self-reported longstanding illness or disability, chest pain at rest, or leg pain while walking". This variable is already adjusted for in analyses (i.e., model 2) to account for its impact, along with other covariates, on the traditional observational associations for PA and LTL. The result, as described, is some attenuation in the magnitude of associations.

REVIEWERS' COMMENTS:

Reviewer #1 (Remarks to the Author):

The authors made most of the revisions I suggested, and the parts that were not made were explained with logical reasons. Furthermore, I see that the suggestions and revisions of the other reviewers were also taken into account by the authors, the revisions were made within the limits of possible, and the necessary logical answers were given to the questions of reviewers. Therefore, I recommend the acceptance of the article.

Reviewer #2 (Remarks to the Author):

The authors have comprehensively addressed my comments/questions. The manuscript adds important results on the relation between walking and LTL.